# Effect of Aprotic Solvents on the Microtensile Bond Strength of Composite Core and Fiber-Reinforced Composite Posts

**DOI:** 10.3390/polym15193984

**Published:** 2023-10-03

**Authors:** Wisarut Prawatvatchara, Somphote Angkanawiriyarak, Awiruth Klaisiri, Tool Sriamporn, Niyom Thamrongananskul

**Affiliations:** 1Department of Prosthodontics, Faculty of Dentistry, Chulalongkorn University, Bangkok 10330, Thailand; 2Faculty of Dentistry, Chulalongkorn University, Bangkok 10330, Thailand; somphote@hotmail.com; 3Division of Restorative Dentistry, Faculty of Dentistry, Thammasat University, Pathumthani 12120, Thailand; dentton@hotmail.com; 4Division of Prosthodontics, College of Dental Medicine, Rangsit University, Pathumthani 12000, Thailand; tool.s@rsu.ac.th

**Keywords:** aprotic solvent, fiber-reinforced composite post, microtensile bond strength, surface treatment

## Abstract

This investigation evaluated the effects of aprotic solvents, i.e., tetrahydrofuran, pyridine, and morpholine, compared with hydrogen peroxide, on the surfaces of fiber-reinforced composite posts with a composite core based on the microtensile bond strength. In total, 150 FRC Postec Plus posts and 150 D.T. Light-Posts were randomly divided into three groups (non-thermocycling, 5000-cycle, and 10,000-cycle thermocycling groups). Each group was divided into five subgroups according to the post-surface treatment: C, non-treatment group; H_2_O_2_, immersed in 35% hydrogen peroxide; THF, immersed in tetrahydrofuran; PY, immersed in pyridine; and MP, immersed in morpholine. The treated specimens were placed in the bottom of a plastic cap and filled with a composite core material in preparation for the microtensile bond test. The data were evaluated using one-way ANOVA and Tukey’s test (*p* < 0.05) as well as an independent t-test (*p* < 0.05). For the surface roughness, white light interferometry was used for measurement, and the mean surface roughness was analyzed via one-way ANOVA and Tukey’s test (*p* < 0.05). The results showed that, under non-thermocycling conditions, the PY subgroup with D.T. Light-Post had the highest microtensile bond strength, followed by THF, MP, H_2_O_2_, and the control groups. For FRC Postec Plus, the PY group had the highest microtensile bond strength, followed by MP, THF, H_2_O_2_, and the control groups. Although the thermocycling conditions decreased the microtensile bond strength in all groups, the PY subgroup still had the highest value. An independent t-test revealed that even under all non-thermocycling and 5000- and 10,000-cycle thermocycling conditions, D.T. Light-Post in the PY subgroup displayed significantly higher microtensile bond strengths than FRC Postec Plus in the PY subgroup. While the surface roughness of the fiber-reinforced composite posts showed that the posts treated with pyridine possessed the highest surface roughness for each material type, In conclusion, as an aprotic solvent, pyridine generates the highest microtensile bond strength between the interfaces of composite cores and fiber-reinforced composite posts.

## 1. Introduction

Dental fiber-reinforced composite posts are widely used in endodontically treated teeth. Because of their similarities with dentin in terms of their elastic moduli [1], these composite posts have been shown to outperform cast metal posts. Abutment buildup around a fiber post is essential when there is considerable loss of coronal tooth structure [2]. Fiber-reinforced composite posts are made from carbon or silica pre-stretched fibers, E-glass (electrical glass), S-glass (high-strength glass), and quartz fibers, which are pure silica in crystalline form and can be found in glass fiber-reinforced composite posts [3,4,5]. The arrangement of the fibers provides tensile strength while enclosing a resin matrix that resists compressive stress [6]. The resin matrix of fiber-reinforced composite posts has a strongly cross-linked structure composed of the aromatic monomer Bis-GMA (bisphenol A glycidyl methacrylate) or epoxy polymers [6,7].

For the core build-up material, resin composite is the material of choice for use with fiber-reinforced composite posts due to its hardness, fracture toughness, and resemblance to the tooth structure, allowing for preparation after curing. When used to perform core restorations, such materials must produce good outcomes in terms of microscopic structural integrity and surface adaptability surrounding the fiber posts [8,9,10,11,12]. The bonding ability of the core material and the fiber post is a key factor influencing the final restoration survival rate [11,12].

Surface treatment of fiber-reinforced composite posts to increase the bond strength between fiber-reinforced composite posts and the composite core interface has been explored in several research studies [6,13,14,15]. In several studies, micro-mechanical and chemical bonding methods were introduced, and sandblasting and hydrofluoric acid etching were employed. However, these techniques compromised the strength of the posts and harmed the structure of the glass fibers. These methods also lacked selectivity and could occasionally damage the post’s structure [16,17,18]. To avoid compromising the integrity of the fiber, agents that dissolve only the epoxy matrix section have been investigated [13,19,20,21]. Chemical reagents such as hydrogen peroxide, potassium permanganate, and sodium ethoxide were utilized because of their capacity to partially dissolve only resin matrices [22]. By removing the epoxy resin’s top layer, a greater surface area of exposed glass or quartz fibers that have hydroxy groups can be made available for silanization. These surface treatments have produced satisfactory results [23]. However, these chemical reagents have significant drawbacks: hydrogen peroxide can be toxic, particularly in high concentrations, and potassium permanganate can impart an undesired color to the surfaces of fiber-reinforced composite posts. In addition, applying potassium permanganate and sodium ethoxide is a more time-consuming process that is not practical in clinical settings [20,21,24].

Aprotic solvents are solvents that are unable to provide hydrogen atoms for protonation. Thus, these solvents neither give nor receive protons, resulting in weak basic properties. Many solvents can be classified as aprotic, including tetrahydrofuran, pyridine, and morpholine. These solvents are frequently employed in the production of paints, coatings, cosmetics, and pharmaceuticals. Although tetrahydrofuran [25,26] and morpholine [27] have been utilized in research on dental applications, pyridine has never been employed for this purpose.

The aim of this investigation was to determine the effects of aprotic solvents, including tetrahydrofuran, pyridine, and morpholine, as well as the conventional chemical reagent hydrogen peroxide, on the surfaces of fiber-reinforced composite posts, considering the surface roughness and microtensile bond strength of composite cores under non-thermocycling, 5000-cycle, and 10,000-cycle thermocycling conditions. The null hypothesis is that aprotic solvents will have no effect on the surface roughness and microtensile bond strength between fiber-reinforced composite posts and composite cores under non-thermocycling and 5000-cycle and 10,000-cycle thermocycling conditions.

## 2. Materials and Methods

### 2.1. Material Preparation

The materials used in this investigation are shown in Table 1.

The sample size calculation was based on a 5% margin of error and 95% confidence level (significance of 0.05) using statistical software (G*Power v. 3.1.5, Faul, Erdfelder, Buchner, and Lang, Heinrich Heine University, Düsseldorf, Germany, http://www.gpower.hhu.de/en.html), accessed on 5 February 2020.

### 2.2. Specimen Preparation

This study used a total of 300 fiber-reinforced composite posts (size 3 maximum diameter) composed of 150 samples each of the FRC Postec Plus and D.T. Light-Post. Each fiber-reinforced composite post group was randomly divided into three groups (non-thermocycling, 5000-cycle, and 10,000-cycle thermocycling groups). Before testing, all specimens were immersed in 24 °C water with an ultrasonic cleaner (Branson5210, Bransonic, CT, USA) for 5 min and stored in a dry place.

### 2.3. Surface Treatment Protocols

Subgroup C: non-etching group, rinsed with deionized water, dried by air blowing, and covered with a silane coupling agent.

Subgroup H_2_O_2_: immersed in 35% hydrogen peroxide for 1 min, rinsed with deionized water, dried by air blowing, and covered with a silane coupling agent.

Subgroup THF: immersed in tetrahydrofuran for 1 min, rinsed with deionized water, dried by air blowing, and covered with a silane coupling agent.

Subgroup PY: immersed in pyridine for 1 min, rinsed with deionized water, dried by air blowing, and covered with a silane coupling agent.

Subgroup MP: immersed in morpholine for 1 min, rinsed with deionized water, dried by air blowing, and covered with a silane coupling agent.

The same technique was used for all subgroups (*n* = 10): the specimens were inserted into a hole in the bottom of the plastic mold, as shown in Figure 1. ExcitE F DSC (Ivoclar Vivadent, Schaan, Liechtenstein) was applied for 10 s with a microbrush to produce only a thin layer, and the excess bonding agent was removed with gentle air drying. Then, the bonding agent was activated using a light-curing unit (Bluephase N^®^, Ivoclar Vivadent, Schaan, Liechtenstein) at high intensity for 20 s.

Multicore flow (Ivoclar Vivadent, Schaan, Liechtenstein) was injected around the post until the plastic mold was fully filled. The post was then light-cured for 40 s per surface.

### 2.4. Microtensile Specimen Preparations

The specimens were stored in a dry environment for 24 h at 37 °C before being sectioned into 1 × 1 mm^2^ cross-sectional bars with a slow-speed diamond machine (Model Isomet, Buehler, IL, USA). The specimens were measured with a digital vernier caliper (Mitutoyo series 500, Mitutoyo, Kawasaki, Japan), as shown in Figure 2.

The thermocycling group was divided based on different chemical agent treatments into five subgroups, just as for the non-thermocycling group.

The thermocycle specimens underwent 10,000-cycle thermocycling between water baths at 5 °C and 55 °C, with a dwell time of 60 s in each water bath.

Each specimen was attached to a metal grip, as shown in Figure 3, which was attached to a Universal Testing Machine (EZ-S 500N, Shimadzu Corporation, Kyoto, Japan) with cyanoacrylate glue (Model Repair ll, Sankin Industry, Tokyo, Japan). Tensile strength was then applied at a crosshead speed of 1 mm/min until failure occurred.

The microtensile bond strength was determined by the applied tension divided by the bonded area. However, because the post-composite core interface was curved, this area was measured using the following mathematical formula:A = 2r arcsin (L/2r) × h 
where r, L, and h are the diameter, width, and thickness of the post, respectively, as shown in Figure 4.

### 2.5. Mode of Failure Evaluation

A stereo microscope (SZ 61, Olympus, Tokyo, Japan) at 40× magnification was used to evaluate and classify the fracture area (adhesive, cohesive, or mixed).

The software ImageJ 1.41 (ImageJ 1.41, Wayne Rasband, National Institutes of Health, Bethesda, MD, USA) was used to measure the surface area. The fracture surface was computed as a percentage of the overall bonding area. When the fiber-reinforced composite posts or composite core material interface were observed in over 60% of the total bonding area, the mode of failure was classified as cohesive failure. If this area was less than 60% but greater than 40%, the materials were classified as having a mixed failure rate. The specimens were characterized as experiencing adhesive failure if these areas comprised less than 40% of the total area.

### 2.6. Sample Preparation for Measuring Surface Roughness

#### 2.6.1. Determination of Surface Properties with White Light Interferometry

For white light interferometry, the 50 specimens from D.T. Light-Post and FRC Postec Plus were employed for surface roughness measurements. Each post type was divided into five subgroups, as follows:

Subgroup C: ten fiber-reinforced composite posts rinsed with deionized water and dried by air blowing.

Subgroup H_2_O_2_: ten fiber-reinforced composite posts immersed in 35% hydrogen peroxide for 1 min, rinsed with deionized water, and dried by air blowing.

Subgroup THF: ten fiber-reinforced composite posts immersed in THF for 1 min, rinsed with deionized water, and dried by air blowing.

Subgroup PY: ten fiber-reinforced composite posts immersed in pyridine for 1 min, rinsed with deionized water, and dried by air blowing.

Subgroup MP: ten fiber-reinforced composite posts immersed in morpholine for 1 min, rinsed with deionized water, and dried by air blowing.


**Sample preparation materials**


The tapered end of a fiber-reinforced composite post in each subgroup was placed 2 mm deep into the plastic mold, as shown in Figure 5.Multicore flow was injected around the post until the plastic mold was fully filled and then activated with a light-curing unit for 40 s per surface.A Rectangular black acrylic resin plate was joined to a fiber-reinforced composite post-plastic mold assembly via a plastic hole that was inserted into the metal post and secured to a post-plastic mold assembly, as shown in Figure 6.The assembled component was placed on the white-light interferometry tester’s stand.The Ra value was measured on the surface of the fiber-reinforced composite posts that corresponded to the marker line. For each subgroup, ten samples were measured. Measurements were made using white light optical interferometry (WLI, ContourX-1000, Bruker, Salbruken, Germany) with the vertical scanning interferometry mode (VSI). A magnification of 20× with a 0.55× field of view (FOV) was used, obtaining an image size of 1.7 × 2.3 mm. The arithmetic mean height of the surface (Ra) was reported.

#### 2.6.2. Scanning Electron Microscope (SEM) Evaluation

Two samples of each type of post that received surface treatments were subjected to SEM for morphological analysis of the surface using a gold coater (Gold sputtering unit, JEOL Ltd., Akishima, Japan) and observed at 500× magnification with a scanning electron microscope (JSM-IT500HR, JEOL Ltd., Tokyo, Japan).

### 2.7. Statistical Analysis

For the non-thermocycling, 5000-cycle, and 10,000-cycle thermocycling conditions, the data from each condition was analyzed using IBM SPSS Statistics for Windows version 22.0. The continuous outcome and normality of distribution of the analyzed data were determined using a Kolmogorov–Smirnov test at a significance level of 0.05. Levene’s test was then used to analyze the equality of variation.

The results showed that the data had a normal distribution and equal variance. Therefore, one-way analysis of variance (ANOVA) was used to analyze the data, followed by Tukey’s Honestly Significant Difference test (*p* < 0.05).

For the same chemical reagent treatment groups, an independent t-test was used to examine statistically significant differences (*p* < 0.05) between the D.T. Light-Post and FRC Postec Plus groups.

In white light interferometry surface roughness measurements, the mean surface roughness values of each type of post exhibited a normal distribution and equal variance. A one-way analysis of variance (ANOVA) was used to analyze the data, followed by Tukey’s Honestly Significant Difference test (*p* < 0.05).

## 3. Results

### 3.1. Mean Microtensile Bond Strength

All the mean microtensile bond strength and standard deviation results in both non-thermocycling conditions and thermocycling conditions are shown in Table 2, Table 3 and Table 4.

#### 3.1.1. Non-Thermocycling Condition

In the epoxy resin matrix groups (D.T. Light-Post), the microtensile bond strength in the PY group was significantly higher than that in the other groups. However, the microtensile bond strength of the MP group was not significantly different from that of THF and H_2_O_2_; however, it was significantly higher than that in the control group.

In the Bis-GMA resin matrix groups (FRC Postec Plus), the microtensile bond strength in the PY group was significantly higher than that in the other groups. The MP group had a significantly higher microtensile bond strength than the THF, H_2_O_2_, and control groups (*p* < 0.05). The microtensile bond strength in the control group did not differ significantly from that in the THF and H_2_O_2_ groups.

Regarding the resin matrix type of fiber-reinforced composite posts, the microtensile bond strength of the epoxy resin matrix post was significantly higher than that of the Bis-GMA resin matrix post in all intervention methods.

#### 3.1.2. The 5000-Cycle and 10,000-Cycle Thermocycling Conditions

For thermocycle testing using 5000-cycle thermocycling, in the epoxy resin matrix groups (D.T. Light-Post), the microtensile bond strength in the PY group was significantly higher than that in the other groups. The microtensile bond strength of the MP group was significantly higher than that of the control group. However, the microtensile bond strength in the control group did not differ significantly from that in the THF and H_2_O_2_ groups.

Under 5000-cycle thermocycling, in the Bis-GMA resin matrix groups (FRC Postec Plus), the PY group had significantly higher microtensile bond strength compared to the other groups. The bond strength in the MP group was also significantly higher than that in the THF and control groups (*p* < 0.05). However, the bond strength in the THF group was not significantly different from that in the H_2_O_2_ and control groups.

Similar to the previous results, under 10,000-cycle thermocycling, in the epoxy resin matrix groups (D.T. Light-Post), the group with the highest microtensile bond strength was the PY group. The bond strength in the MP group was significantly higher than that in the THF, H_2_O_2_, and control groups (*p* < 0.05), while in the Bis-GMA resin matrix groups (FRC Postec Plus), the PY group had significantly higher microtensile bond strength than that in the other groups. The bond strength in the MP group was significantly higher than that in the THF, H_2_O_2_, and control groups (*p* < 0.05). However, the THF, H_2_O_2_, and control groups were not significantly different from each other.

Compared to the resin matrix type of fiber-reinforced composite posts, the Epoxy PY group had significantly higher microtensile bond strength than the Bis-GMA group under both thermocycling conditions.

### 3.2. Failure Modes

The modes of failure for the non-thermocycling group and the 5000- and 10,000-cycle thermocycling groups are presented in Table 5, Table 6 and Table 7 and Figure 7, Figure 8 and Figure 9.

#### 3.2.1. Non-Thermocycling Condition

It was shown that higher adhesive failure was present in all groups, as was mixed failure. Moreover, the highest percentages of mixed failures were observed in the Epoxy PY group. However, cohesive failure was not found in any group.

#### 3.2.2. The 5000-Cycle Thermocycling Condition

A higher adhesive failure mode was discovered in all groups. Mixed failure was also present in all groups except for the Epoxy C, Bis-GMA PY, and Bis-GMA MP groups. However, cohesive failure was evident only in the Epoxy PY and Bis-GMA PY groups.

#### 3.2.3. The 10,000-Cycle Thermocycling Condition

Higher adhesive failure was found in all groups. Mixed failure was also shown in all groups except for the Epoxy H_2_O_2_, Epoxy MP, and Bis-GMA H_2_O_2_ groups. Furthermore, cohesive failure was observed only in the Epoxy PY group, indicating a total failure of 20%.

### 3.3. Surface Roughness Analysis Results

#### 3.3.1. White Light Interferometry

Table 8 and Table 9 present qualitative analyses of roughness parameters using white light interferometry. In the epoxy and Bis-GMA groups, treatment with an aprotic solvent (THF, PY, and MP) resulted in a significant increase in Ra compared with each control group.

#### 3.3.2. Scanning Electron Microscope (SEM)

The SEM results shown in Figure 10 and Figure 11 indicate that all treatments modified the surface topography of the epoxy (D.T. Light-Post) and Bis-GMA resin (FRC Postec Plus) matrix. Pyridine immersion removed both resin matrices significantly more thoroughly than other treatments and exposed more fibers on the surface, as represented by red arrows in Figure 10. In contrast, tetrahydrofuran only slightly removed the Bis-GMA resin matrix, as shown in Figure 11.

## 4. Discussion

This investigation explored the effects of aprotic solvents on the surfaces of fiber-reinforced composite posts, including surface roughness, based on the microtensile bond strength of the composite core under non-thermocycling and 5000- and 10,000-cycle thermocycling conditions.

The results of this investigation revealed that immersion-based fiber-reinforced composite posts using pyridine can significantly increase microtensile bond strength. Therefore, the null hypothesis was rejected. The present study applied the microtensile bond test, which is characterized by using specimens with small bonding areas (less than 2 mm^2^), which decreases the occurrence of defects (flaws) that lower bond strength and increases data variability [28]. Moreover, the microtensile test allows for a more uniform stress distribution than the shear bond strength test due to axial tensile loading on a smaller interface. In this way, the frequency of cohesive fractures is reduced [29]. Furthermore, microtensile testing yields more reliable results and may be used for small samples [30,31]. Several studies found that premature bond failure, which affects the bond strength value, is associated with cutting procedures that induce higher mechanical stress at the interface, in addition to experimental variables such as cutting speed and specimen shape, as well as intrinsic material properties [32,33].

Hydrogen peroxide, an alkaline chemical, partially dissolves the epoxy resin matrix via a substrate oxidation mechanism involving the electrophilic attack of hydrogen peroxide on the cured secondary amine [22,34]. The effects of varying the concentrations and application times of hydrogen peroxide on the fiber posts were investigated. Concentrations of 10% [21], 20% [35], 24% [36], 35% [37,38], and 50% [36] were considered when analyzing the treated posts. However, the application of a 35% concentration over a 1 min duration yielded the highest bond strength [39]. Therefore, the present study employed 35% hydrogen peroxide applied for 1 min via immersion, which was easier to regulate in the experiments, and then applied this strategy to the other surface treatment groups.

Tetrahydrofuran, pyridine, and morpholine were used as aprotic solvents in this investigation. The microtensile bond results revealed that an aprotic solvent can increase the microtensile bond strength more than hydrogen peroxide with the same application time. This suggests that the peroxide’s oxidation effect left residual oxygen on the postsurface [37,38], which could then affect the polymerization process [40], thereby decreasing the microtensile bond strength in the H_2_O_2_ group.

Although the mechanism by which an aprotic solvent functions in the surface treatment of fiber-reinforced composite posts is currently unknown, it is generally accepted that a solvent is suitable for dissolving a polymer if their solubility properties are comparable. The closer the solute and solvent solubility parameters are, the more likely it is that the solute will be soluble in the corresponding solvent [41,42,43]. Therefore, solubility parameters, or numerical values defining the relative behavior of a solvent’s solubility, are one of the most important considerations when selecting a solvent. A dimethacrylate polymer such as Bis-GMA has a solubility parameter of 22.13 MPa^1/2^ [44], while the solubility parameter of epoxy resin is 18.77 MPa^1/2^ [45]. Aprotic solvents have the following solubility parameters: tetrahydrofuran, 18.5 MPa^1/2^; pyridine, 21.7 MPa^1/2^; and morpholine, 22.1 MPa^1/2^ [45]. However, factors other than solubility parameters, such as concentration, temperature, application time, and pH, also influence the efficacy of aprotic solvents [46].

In the present study, we observed a high bond strength in both the non-aprotic and aprotic solvent groups. This could be due to bonding between the bonding agent and the resin matrix as well as between the bonding agent and the fiber-reinforced composite. For quartz or glass fibers, surface treatment of fiber-reinforced composite posts was used to remove the resin matrix on the surface layer, exposing more quartz or glass fibers to produce a chemical bond interaction between silane coupling agents and hydroxy groups on the glass/quartz surface. This chemical bond has a great impact on bond strength.

Notably, the pyridine-treated epoxy resin matrix group had the highest microtensile bond strength, possibly because the pyridine and epoxy resin matrices of glass/quartz fiber posts have a closed solubility parameter. Thus, pyridine can dissolve and swell the epoxy resin matrix, resulting in two processes for improving the bond strength between fiber-reinforced composite posts and the composite core build-up material:
The resin matrix covering the glass fiber was dissolved and removed, exposing the glass fiber posts. Furthermore, the fibers contained a hydroxyl group, which could be silanated to create sufficient adhesion to the bonding agent.When the resin matrix was dissolved and swelled, adhesion between the polymer matrix and the new bonding resin could be achieved via two processes:
(i)Mechanical interlocking is caused by rough surfaces.(ii)Adhesion via interpenetrating network (IPN) formation. When the resin matrix swelled, became soft, and created pores between the polymer chains, these processes permitted the transfer of low-molecular-weight monomers from the bonding agent into the swollen polymer matrix via diffusion.

In this study, we chose white light interferometry for surface roughness measurements because this technique is a non-contact optical method for developing 3D profiles of rough and smooth surfaces [47,48,49]. Since the surface roughness values varied depending on the area of the fiber-reinforced composite post that was analyzed, we decided to use a black acrylic plate with metal posts and marker lines to reposition the specimens in order to measure roughness in the same areas as the fiber-reinforced composite posts. In this way, we obtained more reliable results.

For the SEM investigation, the results demonstrated changes in surface topography in all surface-treated groups. The pyridine groups presented more exposed glass fibers than the control group, which confirmed the results of white light interferometry, indicating the highest roughness and surface morphology values.

In terms of failure mode, we observed no cohesive failure in the non-thermocycling groups; only the PY groups exhibited an increase in cohesive failure after 5000 and 10,000 cycles of thermocycling, which correlated with the microtensile bond strength of the pyridine groups, which had the highest bond strength under both thermocycling conditions.

There was also a relationship between thermocycling and the mode of failure. Thermocycling is used to weaken a material by aging the bonding interface and resin composite [50]. In the present study, 5000- and 10,000-cycle thermocycling at 5 °C and 55 °C was used to simulate a six-month and one-year period, respectively, in an oral environment [51]. In the 5000- and 10,000-cycle thermocycling groups, all cohesive failures occurred at the post interface. These results agree with a previous study that reported the mechanical properties of a fiber-reinforced composite post to decrease after thermocycling. Thus, the weakest area was located at the fiber-reinforced composite post interface, as indicated by cohesive failure.

After being exposed to thermocycling, differences in the coefficients of thermal expansion (CTE) for each component could influence a decrease in microtensile bond strength [52,53]. Moreover, void spaces in the resin matrix could promote water absorption and result in decreased strength of the fiber-reinforced composite post. 

The advantages of these discoveries could be used in dental practices to enhance the bond strength of composite cores and fiber-reinforced composite posts in the case of endodontically treated teeth.

For future studies, the specimens will be soaked in the solution for one minute without being washed with water. We anticipate that by using this reduction technique, chair time and working steps will be reduced without affecting the bond strength between the composite core and the fiber-reinforced composite post.

## 5. Conclusions

Based on the limitations of this study, the following can be concluded:(1)Pyridine, an aprotic solvent, generates the highest microtensile bond strength between the interfaces of composite cores and fiber-reinforced composite posts, e.g., Bis-GMA (41.09 ± 4.29 MPa) and an epoxy resin matrix (46.93 ± 4.97 MPa). However, pyridine is more effective with an epoxy resin matrix than with a Bis-GMA resin matrix.(2)In this research, pyridine treatment resulted in the highest surface roughness (Epoxy 704.140 ± 44.637 Ra, Bis-GMA 859.439 ± 30.789 Ra).(3)Thermocycling reduced the microtensile bond strength in all groups. However, the pyridine groups had the highest bond strength.

## Figures and Tables

**Figure 1 polymers-15-03984-f001:**
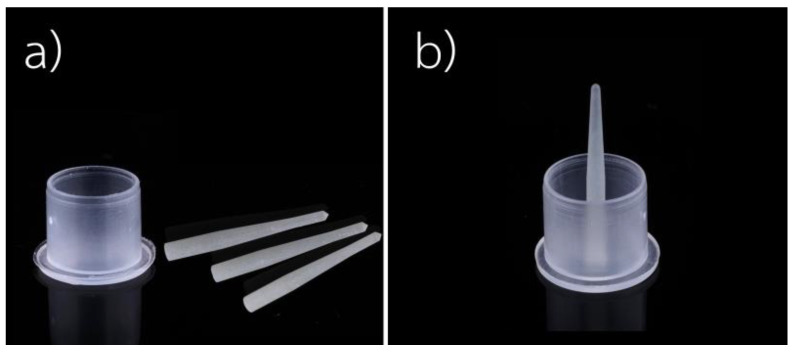
(**a**) A plastic cap with a hole at the bottom and fiber-reinforced composite posts. (**b**) A fiber-reinforced composite post is inserted into the hole.

**Figure 2 polymers-15-03984-f002:**
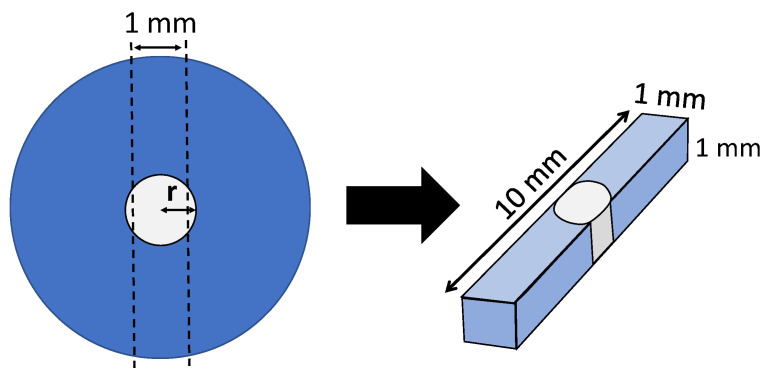
Cross-section of bar-shaped specimen.

**Figure 3 polymers-15-03984-f003:**
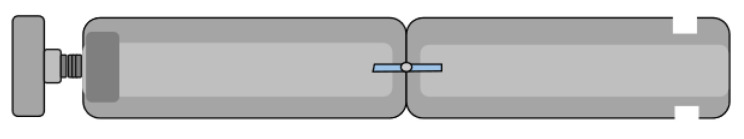
Schematic representing the specimen attached to a metal grip.

**Figure 4 polymers-15-03984-f004:**
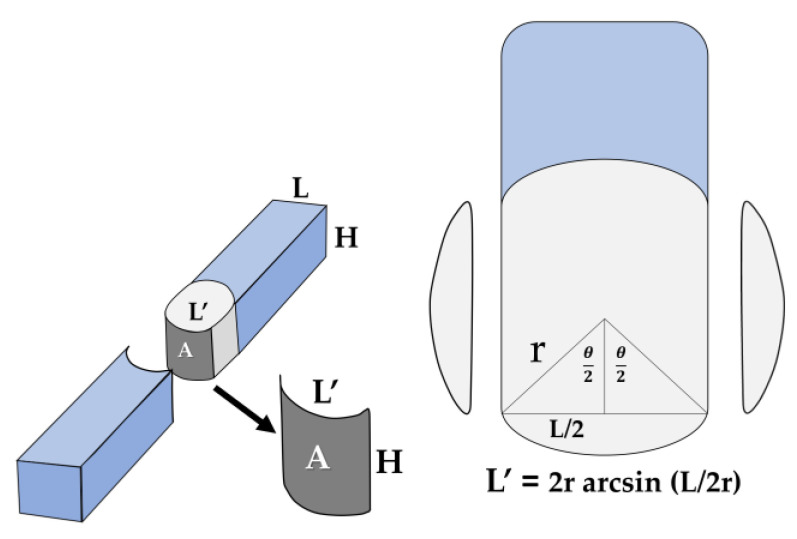
Graphic illustration of bond area calculations.

**Figure 5 polymers-15-03984-f005:**
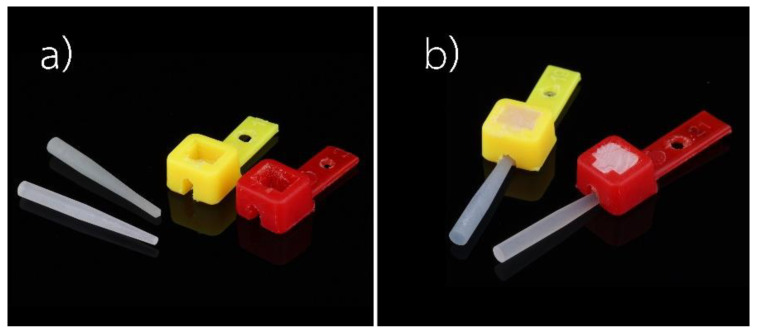
(**a**) Fiber-reinforced composite posts and a plastic mold with a hole in the holder; (**b**) insertion and fixation of the tapered end of fiber-reinforced composite posts into the plastic molds.

**Figure 6 polymers-15-03984-f006:**
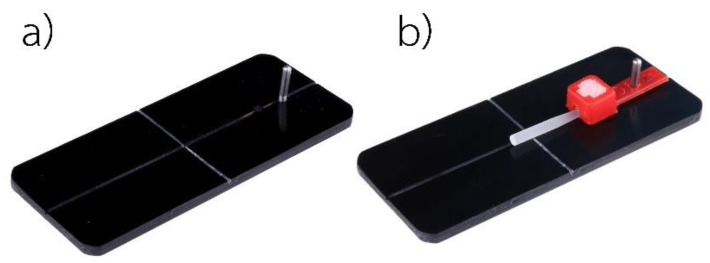
(**a**) A rectangle-shaped black acrylic resin plate with marker lines and a short metal post; (**b**) a fiber-reinforced composite post-plastic mold assembly connected on top of a black acrylic resin plate.

**Figure 7 polymers-15-03984-f007:**
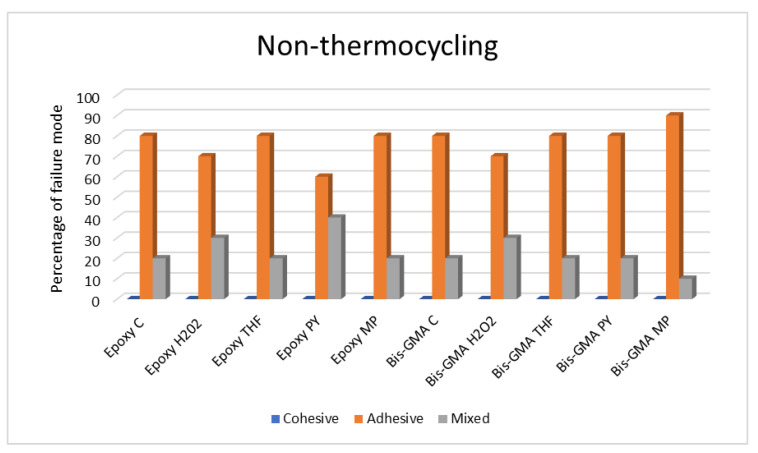
Bar graph of failure modes in the absence of thermocycling.

**Figure 8 polymers-15-03984-f008:**
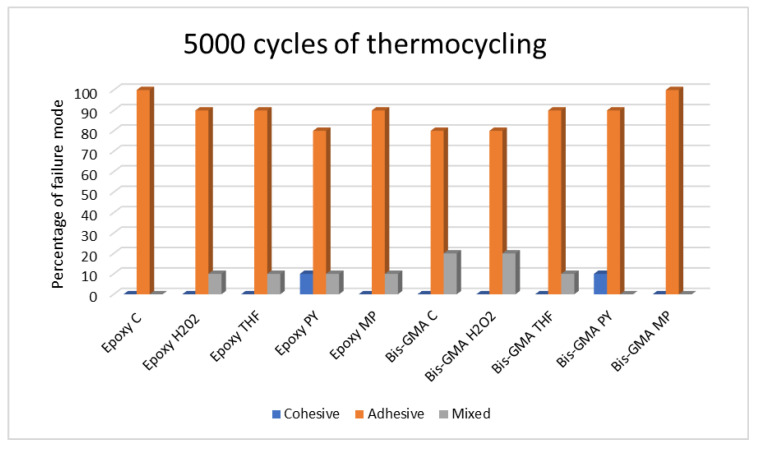
Bar graph of failure modes after 5000 thermocycling cycles.

**Figure 9 polymers-15-03984-f009:**
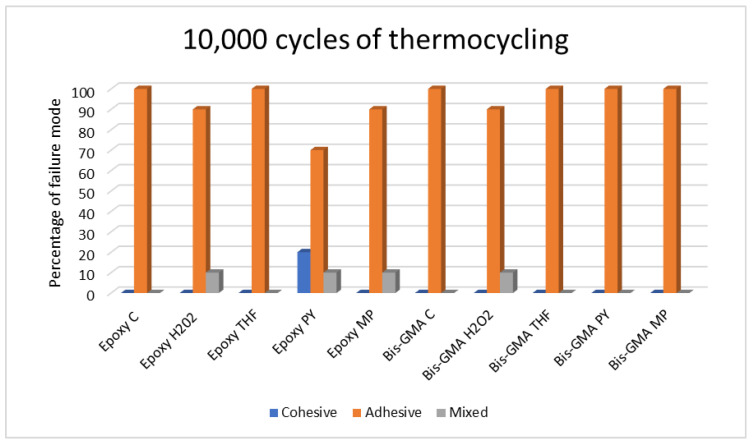
Bar graph of failure modes after 10,000-cycle thermocycling.

**Figure 10 polymers-15-03984-f010:**
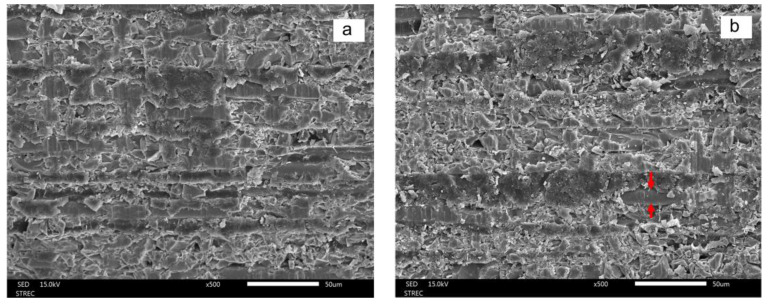
SEM images (×500) showing the surface topography of the epoxy (D.T. Light-Post) control (**a**) and treatment groups, which include hydrogen peroxide (**b**), tetrahydrofuran (**c**), pyridine (**d**), and morpholine (**e**), after 1 min of immersion. Note the exposed glass fibers (red arrows).

**Figure 11 polymers-15-03984-f011:**
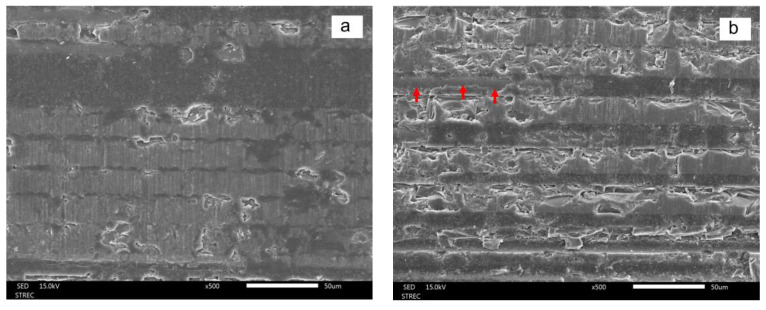
SEM images (×500) showing the surface topography of the Bis-GMA (FRC Postec Plus) control (**a**) and treatment groups, which include hydrogen peroxide (**b**), tetrahydrofuran (**c**), pyridine (**d**), and morpholine (**e**), after 1 min of immersion. Note the exposed quartz fibers (red arrows).

**Table 1 polymers-15-03984-t001:** Trade names, manufacturers, and compositions of experimental materials.

Type	Material	Composition
Fiber-reinforced composite post	FRC Postec Plus (Ivoclar Vivadent, Schaan, Liechtenstein)	Glass fiber, Bis-EMA ^1^, ytterbium trifluoride, Bis-GMA ^2^, 1,4-butanediol dimethacrylate
Fiber-reinforced composite post	D.T. Light-Post (Recherches Techniques Dentaires, RTD, St. Egreve, France)	Quartz fibers, epoxy resin matrix
Composite core	Multicore Flow (Ivoclar Vivadent, Schaan, Liechtenstein)	Matrix: Bis-GMA ^2^, urethane dimethacrylate, triethylene glycol dimethacrylate Fillers: barium glass, ytterbium Ba-Al-fluorosilicate glass, highly dispersed silicon dioxide
Adhesive agent	ExcitE F DSC (Ivoclar Vivadent, Schaan, Liechtenstein)	Bis-GMA ^2^, ethanol, 2-hydroxyethyl methacrylate phosphonic acid acrylate, diphenyl (2,4,6-trimethylbenzoyl) phosphine oxide, potassium fluoride
Silane coupling agent	Monobond plus (Ivoclar Vivadent, Schaan, Liechtenstein)	Ethanol, 3-(trimethoxysilyl)propyl methacrylate, methacrylated phosphoric acid ester
Treatment reagents	Tetrahydrofuran (LOBA Chemie Pvt. Ltd.—Mumbai, India)	Tetrahydrofuran 99.8%
Treatment reagents	Hydrogen peroxide (LOBA Chemie Pvt. Ltd.—Mumbai, India)	Hydrogen peroxide 35%
Treatment reagents	Pyridine (LOBA Chemie Pvt. Ltd.—Mumbai, India)	Pyridine 99.5%
Treatment reagents	Morpholine (LOBA Chemie Pvt. Ltd.—Mumbai, India)	Morpholine 99.5%

^1^ Abbreviations: Bis-EMA, bisphenol A polyethylene glycol diether dimethacrylate; ^2^ Abbreviations: Bis-GMA, bisphenol A-glycidyl methacrylate.

**Table 2 polymers-15-03984-t002:** Mean microtensile bond strength (MPa) resulting from non-thermocycling conditions.

Group	C	H_2_O_2_	THF	PY	MP
Epoxy (D.T. Light-Post)	33.22 ±2.43 ^c^*	38.3 ± 3.16 ^b^*	40.73± 3.40 ^b^*	46.93 ± 4.97 ^a^*	39.40 ± 2.80 ^b^*
Bis-GMA (FRC Postec Plus)	29.64 ± 2.26 ^c^	31.15 ± 2.61 ^bc^	28.49 ± 1.86 ^c^	41.09 ± 4.29 ^a^	34.44 ± 3.76 ^b^

Identical letters in each row (horizontally) indicate no statistically significant differences (*p* > 0.05), while non-identical letters in each row indicate statistically significant differences (*p* < 0.05). The asterisk (*) in each column (vertically) indicates statistically significant differences (*p* < 0.05) between FRC Postec Plus and D.T. Light-Post.

**Table 3 polymers-15-03984-t003:** Mean microtensile bond strength (MPa) resulting from 5000-cycle thermocycling.

Group	C	H_2_O_2_	THF	PY	MP
Epoxy (D.T. Light-Post)	27.81 ± 3.23 ^c^	29.43 ± 3.58 ^bc^	29.73 ± 4.60 ^bc^	41.96 ± 4.45 ^a^*	34.80 ± 5.07 ^b^
Bis-GMA (FRC Postec Plus)	25.05 ± 4.71 ^c^	26.19 ± 2.36 ^bc^	26.31 ± 4.51 ^c^	36.21 ± 5.39 ^a^	33.04 ± 4.43 ^b^

Identical letters in each row (horizontally) indicate no statistically significant differences (*p* > 0.05), while non-identical letters in each row indicate statistically significant differences (*p* < 0.05). The asterisk (*) in each column (vertically) indicates statistically significant differences (*p* < 0.05) between FRC Postec Plus and D.T. Light-Post.

**Table 4 polymers-15-03984-t004:** Mean microtensile bond strength (MPa) mode of failure resulting from 10,000-cycle thermocycling.

Group	C	H_2_O_2_	THF	PY	MP
Epoxy (D.T. Light-Post)	26.33 ± 2.51 ^c^	28.84 ± 3.86 ^bc^	28.59 ± 3.17 ^bc^	39.76 ± 2.37 ^a^*	31.42 ± 4.15 ^b^
Bis-GMA (FRC Postec Plus)	24.22 ± 4.07 ^c^	25.40 ± 2.18 ^c^	25.59 ± 2.51 ^c^	35.26 ± 3.65 ^a^	31.63 ± 4.97 ^b^

Identical letters in each row (horizontally) indicate no statistically significant differences (*p* > 0.05), while non-identical letters in each row indicate statistically significant differences (*p* < 0.05). The asterisk (*) in each column (vertically) indicates statistically significant differences (*p* < 0.05) between FRC Postec Plus and D.T. Light-Post.

**Table 5 polymers-15-03984-t005:** Failure modes under the non-thermocycling condition.

Group	Adhesive	Mode of Failure Mixed	Cohesive
Epoxy C	80	20	0
Epoxy H_2_O_2_	70	30	0
Epoxy THF	80	20	0
Epoxy PY	60	40	0
Epoxy MP	80	20	0
Bis-GMA C	80	20	0
Bis-GMA H_2_O_2_	70	30	0
Bis-GMA THF	80	20	0
Bis-GMA PY	80	20	0
Bis-GMA MP	90	10	0

**Table 6 polymers-15-03984-t006:** Failure modes under the 5000-cycle thermocycling condition.

Group	Adhesive	Mode of Failure Mixed	Cohesive
Epoxy C	100	0	0
Epoxy H_2_O_2_	90	10	0
Epoxy THF	90	10	0
Epoxy PY	80	10	10
Epoxy MP	90	10	0
Bis-GMA C	80	20	0
Bis-GMA H_2_O_2_	80	20	0
Bis-GMA THF	90	10	0
Bis-GMA PY	90	0	10
Bis-GMA MP	100	0	0

**Table 7 polymers-15-03984-t007:** Failure modes under the 10,000-cycle thermocycling condition.

Group	Adhesive	Mode of Failure Mixed	Cohesive
Epoxy C	100	0	0
Epoxy H_2_O_2_	90	10	0
Epoxy THF	100	0	0
Epoxy PY	70	10	20
Epoxy MP	90	10	0
Bis-GMA C	100	0	0
Bis-GMA H_2_O_2_	90	10	0
Bis-GMA THF	100	0	0
Bis-GMA PY	100	0	0
Bis-GMA MP	100	0	0

**Table 8 polymers-15-03984-t008:** Surface roughness (Ra) in the epoxy groups was determined using white light interferometry.

Group	Ra (µm)
Epoxy C	514.860 ± 31.801 ^c^
Epoxy H_2_O_2_	596.661 ± 32.910 ^b^
Epoxy THF	614.874 ± 36.120 ^b^
Epoxy PY	704.140 ± 44.637 ^a^
Epoxy MP	664.629 ± 37.545 ^ab^

Identical letters in the column indicate no statistically significant differences (*p* > 0.05), while non-identical letters in the column indicate statistically significant differences (*p* < 0.05).

**Table 9 polymers-15-03984-t009:** Surface roughness (Ra) in the Bis-GMA groups was determined using white light interferometry.

Group	Ra (µm)
Bis-GMA C	584.970 ± 22.221 ^b^
Bis-GMA H_2_O_2_	585.884 ± 31.036 ^b^
Bis-GMA THF	608.670 ± 36.106 ^b^
Bis-GMA PY	859.439 ± 30.789 ^a^
Bis-GMA MP	621.812 ± 41.610 ^b^

Identical letters in the column indicate no statistically significant differences (*p* > 0.05), while non-identical letters in the column indicate statistically significant differences (*p* < 0.05).

## Data Availability

Not applicable.

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
