# Peer review of "Effect of Aprotic Solvents on the Microtensile Bond Strength of Composite Core and Fiber-Reinforced Composite Posts"

_polymers, 2023, doi:10.3390/polym15193984_

Round 1
Reviewer 1 Report
The article submitted by Prawatvatchara et al., entitled “Effect of aprotic solvents on the microtensile bond strength of composite cores and fiber-reinforced composite posts” (polymers-2568014), investigated the effects of aprotic solvents on the surface of fiber-reinforced composites posts. The results showed that the surface treatment using the different aprotic solvents had different effects on the bonding between the composite posts and composite cores. The topic is interesting. However, this article is so difficult to read that the reviewer has spent much time to guess what the authors want to express. The language of this article should be carefully edited. For the benefit of the reader, the following concerns should be carefully considered.
1) Sentences and words that are difficult to read and understand:
L17: D.T. light …
L82: on the surface of fiber-reinforced composite posts on surface roughness and the microtensile bond strength of composite core under non-thermocycling,
L100: size NO 3
L118: displayed in Figure 18
L118: ExcitE F DSC
L136: same Non thermocycling group
L156: Therefore, …
L208: a fiber-reinforced composite posts-plastic mold assembly connected on top of the black acrylic resin plate.
L303: But cohesive failure was not found in any group.
L344, Therefore, …
L439, From the limitations of this study, …
… …
The above are just some examples, in fact, the language and writing style of this article is needed to be improved thoroughly.
2) In figure 10 and figure 11, the author should mark what they want to express, not only give the figures in their original style.
3) Authors should discuss the mechanism of improving the bonding caused by the surface treatment, not only introduce the experimental results. For example, authors mentioned, “Adhesion by interpenetrating network (IPN) formation”, why can you judge the IPN formation?
In conclusion, a major revision is suggested.
This article is so difficult to read that the reviewer has spent much time to guess what the authors want to express. The language of this article should be carefully edited. The article needed to be re-written.
Author Response
Thank you for valuable comments and suggestions and giving us the opportunity to submit a revised draft of our manuscript. The following changes have been explaining in attachment file.

Reviewer 2 Report
In this paper, the authors investigate the effect of aprotic solvents and the conventional chemical reagent, on the surface of fiber-reinforced composite posts, on surface roughness and the microtensile bond strength of composite core.
From my point of view there are some aspects to improve:
1. Abstract should contain a short introduction about the main subject. Abstract does not contain any quantitative data about the results. Abstract should be revised.
2.All the notations must be explained at the first used.
3. The following paragraph is unclear: “A total of 300 fiber-reinforced composite posts that had a maximum diameter (size No 3) composed of 150 samples each of FRC post plus and D.T. light- post.” Please explain it clearly. What does mean “size No 3”.
4.How many samples were used for each subgroup? Line 117 “The same technique was used for all subgroups…”
5. The shape and the size of samples are unclear. Please explain them.
6. Please explain the component from the Figure 1.
7. “The same technique was used for all subgroups: The specimens were inserted into the hole in bottom of plastic mold as displayed in Figure 18”. Please check the numbering of Figures.
8. All the images from Figure 10 and 11 should be explained. All the SEM should be explained in text and indicate on figure by arrows the characteristic components of the microstructure
9. “one-way analysis of variance (ANOVA) was used to analyze the data” but the results of this analysis are missing. Anova table and some graphical representation of the results are required. Also, what are the value of p and F indicators?
10. The values obtained for surface roughness Ra (between 500 micron to 900 micron) seem to be too high. How do you explain these increased values?
11. How was verified the repeatability of the data?
12. The conclusion section must be extremely specific in terms of obtaining the behavior, results or possibly in numeric values.
13. Please explain what is new in this study. Also the practical application of this study should be mentioned.
Moderate editing of English language required
Author Response
Thank you for your valuable comments and suggestion and thank you for giving us the opportunity to submit a revused draft of our manuscript

Round 2
Reviewer 1 Report
The revised version is better than before. However, the abstract must be rewritten. It is needed to be shortened.
Author Response
I would like to express my gratitude for insightful replies and suggestion you provided. We are grateful for all comments and revisions you provided to newly revised manuscript.
Respectfully
Wisarut Prawatvatchara
